# Scale-Aware Domain Harmonization for Domain Adaptation Person Search

**Huibing Wang** [1]  **Guojian Zhao** [1]  **Jinjia Peng** [2]  **Linfeng Qi**[* 1]  **Mingze Yao**[* 1]  **Jiqing Zhang**[* 1]

## Abstract

Unsupervised Domain Adaptation (UDA) person search aims to transfer a model trained on a labeled source domain to an unlabeled target domain without using target annotations. However, existing UDA methods frequently neglect the issue of scale inconsistency between the source and target domains. This inconsistency arises from variations in camera height, tilt angle, focal length, and scene layout. To address this challenge, we propose a Scale-Aware Consistent Alignment Learning (SCALE) framework. Specifically, we propose a Scale-aware Domain Harmonization (SDH) module, which adaptively harmonizes semantic and structural scales through cross-path interaction and consistency refinement to alleviate cross-domain scale inconsistency. To further reduce pseudo-label noise, we introduce a Bidirectional Cluster Regularization (BCR) strategy, which improves pseudo-label reliability by refining the clustering results through a second regularized clustering step. By collaboratively alleviating the impact of scale misalignment and enhancing pseudo-label reliability, our approach achieves state-of-the-art performance on two benchmark person search datasets, with 82.3% mAP and 84.0% top-1 on the CUHK-SYSU dataset, 41.7% mAP and 82.4% top-1 on the PRW dataset. Our source code is available at https://github.com/whhbdmu/SCALE.

## 1. Introduction

Person search aims to simultaneously detect and identify specific individuals within open real-world scenes. Essen-

*Corresponding authors. [1] School of Information Science and Technology, Dalian Maritime University, Dalian, China [2] School of Cyber Security and Computer, Hebei University, Baoding, China . Correspondence to: Linfeng Qi <qilinfeng@dlmu.edu.cn>, Mingze Yao <ymz0284@dlmu.edu.cn>, Jiqing Zhang <jqz@dlmu.edu.cn>.

*Proceedings of the 43rd International Conference on Machine Learning*, Seoul, South Korea. PMLR 306, 2026. Copyright 2026 by the author(s).

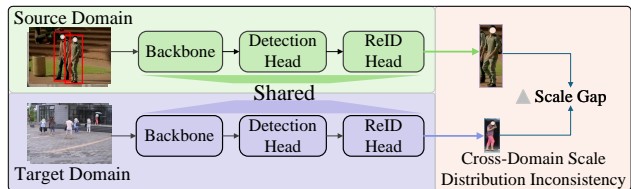

*(a)* Existing Person Search Methods

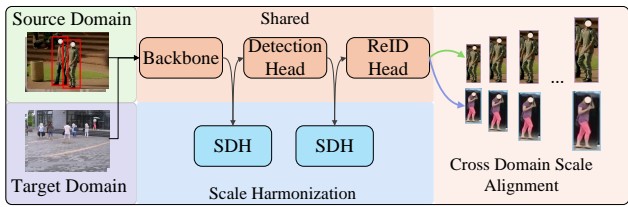

*(b)* Our Proposed SCALE Method

*Figure 1.* (a) Existing methods directly share cross-domain features while overlooking significant scale differences between source and target domains, resulting in a pronounced feature gap. (b) Our proposed approach explicitly harmonizes scales across domains, aligning their scale distributions and substantially narrowing the feature gap to produce more reliable pseudo-labels.

tially, it integrates detection (Girshick et al., 2014; Ren et al., 2015) and re-identification (Re-ID) (Ye et al., 2020) into a unified framework. Most existing methods (Cai et al., 2025; Jiang et al., 2024; Li & Miao, 2021; Xiao et al., 2017; Chen et al., 2020) adopt the supervised learning paradigm, where both branches are jointly optimized on large-scale annotated datasets (Li & Miao, 2021; Jiang et al., 2024). Although these approaches achieve remarkable success within a single domain, they tend to decline effectiveness when transferred to unseen domains, due to the inherent discrepancies in illumination, viewpoints, camera configurations, and background contexts. Such domain sensitivity severely limits the deployment of person search systems in practical cross-scene scenarios.

Unsupervised Domain Adaptation (UDA) (Long et al., 2016; Sun et al., 2019) has recently drawn increasing attention in person search, as it enables transferring discriminative knowledge from a labeled source domain to an unlabeled target domain without requiring additional annotations (Ganin & Lempitsky, 2015; Kang et al., 2019). Current UDA methods (Li et al., 2022; Qi et al., 2025; Cui et al., 2024; Almansoori et al., 2024; Peng et al., 2026; Tan et al., 2026; Peng et al., 2023) employ source-trained models to pro-

duce target-domain pseudo labels (Lee et al., 2013) through clustering, which are subsequently used to iteratively adapt feature representations in conjunction with source-domain data. Following this strategy, Li et al. (Li et al., 2022) first introduced the DAPS framework, which employs implicit domain alignment to learn domain-invariant embeddings. Building upon DAPS (Li et al., 2022), DDAM (Almansoori et al., 2024) generates mixed-domain representations to further enhance adaptability. DSCA (Qi et al., 2025) introduces a dual self-calibration mechanism that jointly refines image and instance level features for more robust pseudo supervision. Meanwhile, recent study (Yao et al., 2025) on incomplete multi-view clustering suggest that recovering structural consistency from incomplete or noisy representations can substantially improve clustering robustness and label reliability.

Despite the remarkable success achieved by existing methods, the majority of them primarily focus on appearance alignment or pseudo-label refinement, while overlooking cross-domain scale inconsistency. As shown in Fig. 1, diagram (a) depicts current approaches that ignore scale disparities between domains, leading to a persistent scale gap in shared feature representations. In contrast, diagram (b) illustrates our proposed solution, which explicitly performs cross-domain scale harmonization, thereby narrowing the feature gap and producing consistent, domain-aligned representations.

To address this issue, we propose a Scale-Aware Consistent Alignment Learning (SCALE) framework for robust unsupervised domain adaptive person search. At its core, the Scale-aware Domain Harmonization (SDH) module adaptively calibrates both semantic and structural scales of multi-resolution features across domains. Specifically, SDH models the hierarchical interactions between spatial structures and semantic representations through a dual-path design, enabling the network to dynamically align feature distributions under inconsistent resolutions and diverse camera conditions. This adaptive harmonization effectively mitigates the semantic bias caused by mismatched detection resolutions and varying pedestrian sizes, ensuring scale-consistent feature representations across domains. In parallel, we introduce a Bidirectional Cluster Regularization (BCR) strategy, which enhances pseudo-label reliability through adaptive split–merge regularization. By jointly harmonizing multi-scale representations and refining clustering consistency, our method effectively mitigates cross-domain scale discrepancies and enhances the robustness and reliability of the model generalization. Extensive experiments on CUHK-SYSU and PRW demonstrate that our method achieves state-of-the-art performance, significantly outperforming existing UDA person search methods under cross-domain scale variation.

In summary, this work makes the following contributions:

- This paper proposes SCALE, a novel dual-stage framework for unsupervised domain-adaptive person search that effectively mitigates cross-domain scale discrepancies and promotes consistent feature learning across domains.
- To align multi-resolution features across domains, a Scale-aware Domain Harmonization (SDH) module is proposed, generating scale-invariant and domain-consistent embeddings.
- We introduce a Bidirectional Cluster Regularization (BCR) strategy to improve pseudo-label reliability via adaptive split–merge regularization, ensuring stable clustering and more robust identity learning.

## 2. Method

In this section, we present the Scale-Aware Consistent Alignment Learning (SCALE) framework, which jointly tackles cross-domain scale inconsistency and pseudo-label unreliability. We first overview the framework, then detail the Scale-aware Domain Harmonization (SDH) module for semantic-scale alignment and the Bidirectional Cluster Regularization (BCR) strategy for clustering stabilization.

### 2.1. Framework overview

Our method is built upon the DSCA (Qi et al., 2025) and extends it into a dual-stage adaptation framework designed to jointly mitigate cross-domain scale inconsistency and pseudo-label unreliability. As illustrated in Fig. 2, the proposed SCALE framework consists of two complementary stages: a Clustering stage with Bidirectional Cluster Regularization (BCR) and a Training stage with Scale-aware Domain Harmonization (SDH). These two components operate in an iterative manner to progressively enhance feature alignment and label stability during unsupervised domain adaptation. In the clustering stage, instance features extracted in inference mode are first used to perform initial clustering. Based on the preliminary clusters, the proposed Bidirectional Cluster Regularization (BCR) further optimizes the distance matrix to produce more reliable pseudo labels. A second clustering is then conducted using the refined distances, and the resulting pseudo labels are employed for target-domain training. During training, the model is first pre-trained on the source domain in a fully supervised manner. The source-trained model is then used to extract image features from the target domain, where pseudo labels from the clustering stage serve as supervision. Region proposals are generated by the RPN and directly aligned via ROI-Align. Finally, both the detection head and the ReID head, equipped with the cross-domain scale-aware SDH module, are jointly optimized to adapt the network parameters for domain-invariant person search.

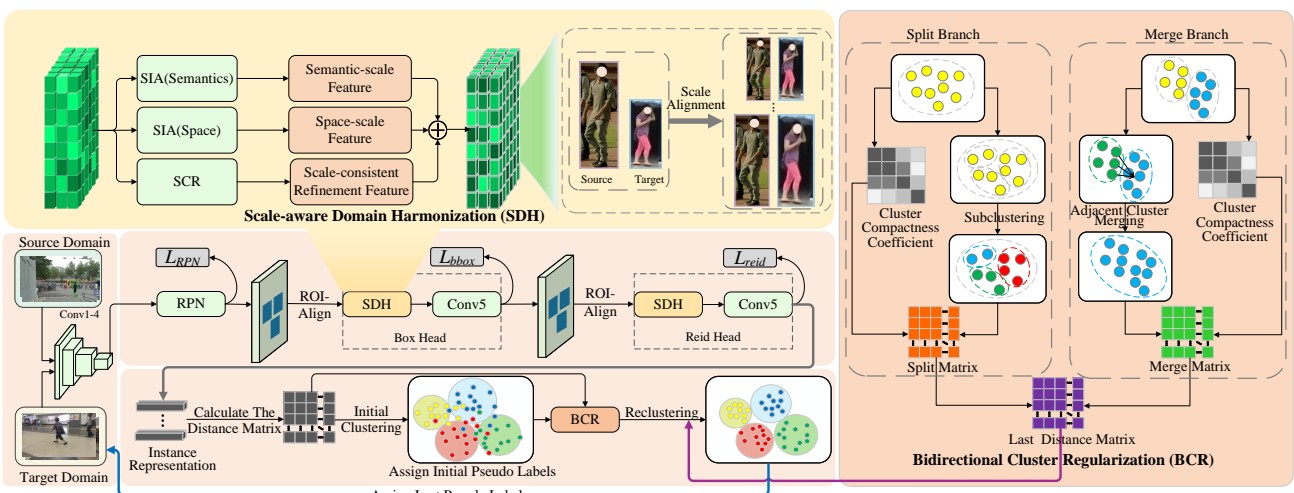

*Figure 2.* Overview of the proposed SCALE framework for unsupervised domain adaptive person search. The framework consists of two stages: (a) Clustering Stage and (b) Training Stage. In the clustering stage, instance features extracted from the target domain are first grouped via initial clustering. The proposed Bidirectional Cluster Regularization (BCR) then refines the distance matrix through adaptive split–merge operations, yielding more reliable pseudo labels for subsequent training. In the training stage, the model is pre-trained on the labeled source domain and then adapted to the target domain using pseudo-labeled data. The Scale-aware Domain Harmonization (SDH) module, embedded in both detection and ReID head, adaptively calibrates semantic and structural scales across domains, ensuring consistent feature alignment under cross-domain scale variations.

## 2.2. Scale-Aware Domain Harmonization (SDH)

Cross-domain person search faces inherent scale inconsistency, arising from variations in pedestrian distances, camera viewpoints, and environmental conditions across scenes, which lead to inconsistent visual scales. These variations introduce semantic bias in both detection and re-identification branches, resulting in unstable domain alignment. To address this challenge, we propose a Scale-Aware Domain Harmonization (SDH) module that mitigates cross-domain scale inconsistency by jointly modeling semantic and structural scales through hierarchical feature interaction, thereby establishing a unified and scale-consistent representation space. The structure of SDH is shown in the Fig. 3 below. It operates through two complementary modules: a Scale-Interactive Alignment (SIA) module and a Scale Consistency Refinement (SCR) module.

**Scale-Interactive Alignment (SIA).** SIA serves as an initial calibration module that explicitly models multi-level feature interactions between spatial and semantic representations through a dual-path design. Given an input feature map $X \in \mathbb{R}^{C \times H \times W}$, SIA contains two parallel branches: a spatial interaction path and a semantic interaction path, both built upon directional attention modeling and multi-scale aggregation.

**Spatial interaction path.** To alleviate cross-domain scale inconsistency that arises from mismatched resolutions and viewpoint variations, we design a spatial alignment pathway that integrates tri-directional attention with multi-scale

contextual fusion. Traditional single-view convolutional encoding often fails to maintain geometric consistency when pedestrian scales vary across domains, leading to spatially distorted activations. In contrast, our tri-directional attention explicitly models feature dependencies along orthogonal planes and captures complementary cues from the height, width, and channel dimensions, thus enabling adaptive recalibration of spatial responses under varying observation scales.

For each projection plane, a direction-aware attention operator recalibrates feature responses by exploiting context along orthogonal dimensions. Formally, the attention operation is defined as:

$$\mathcal{A}_{(\cdot)}(X) = X \odot \sigma\big(\text{BN}(\text{Conv}(Z))\big), \quad (1)$$

where $\odot$ denotes element-wise multiplication, $\sigma(\cdot)$ is the sigmoid gating, and $\text{BN}(\cdot)$ and $\text{Conv}(\cdot)$ denote batch normalization and convolutional transformation, respectively. The input $Z$ concatenates spatial statistics obtained by max and average pooling:

$$Z = [\,\text{MaxPool}(X),\ \text{AvgPool}(X)\,], \quad (2)$$

which captures both strong and smooth activation patterns to guide attention generation.

Based on these directional operators, the tri-directional interaction is formulated as:

$$\mathcal{T}_{\text{sp}}(X) = \frac{1}{3}\big(\mathcal{A}_{\text{cw}}(X^{(C,W)}) + \mathcal{A}_{\text{hc}}(X^{(H,C)}) + \mathcal{A}_{\text{hw}}(X^{(H,W)})\big), \quad (3)$$

where $\mathcal{T}_{sp}(X)$ denotes the aggregated spatial interaction feature, and $\mathcal{A}_{cw}$, $\mathcal{A}_{hc}$, and $\mathcal{A}_{hw}$ correspond to attention operations applied on the channel–width, height–channel, and height–width planes, respectively. By jointly capturing dependencies across three orthogonal projections, this mechanism enables each spatial element to be adaptively refined by its complementary directional context, thus enhancing geometric consistency and mitigating cross-domain scale discrepancies.

To further enhance scale adaptivity, we employ a multi-scale convolutional fusion:

$$\mathcal{M}_{sp}(X) = \delta\Big(\text{BN}\big([\text{DWConv}_k(X)]_{k \in \{1,3,5,7\}}\big)\Big), \quad (4)$$

where $\delta(\cdot)$ denotes ReLU activation and $[\cdot]$ represents channel concatenation. This multi-scale fusion captures contextual cues from different receptive fields, helping preserve localization-sensitive geometric semantics.

$$F_{sp} = \mathcal{M}_{sp}(\mathcal{T}_{sp}(X)), \quad (5)$$

where $\mathcal{F}_{sp}$ denotes spatial path feature. By aggregating directional context along multiple spatial dimensions, this path adaptively aligns structural information across different spatial scales. As a result, it mitigates geometric distortions and preserves consistent spatial responses under cross-domain scale variations.

**Semantic interaction path.** Although the spatial path already captures certain semantic cues through its cross-directional attention design, the resulting semantic information remains implicit and spatially entangled. This is because the spatial aggregation mainly focuses on structural alignment, where semantic relationships are only indirectly reflected through spatial correlations. However, in cross-domain scenarios, domain gaps often manifest not only as spatial distortions but also as semantic-scale discrepancies caused by differences in appearance granularity and identity-level abstraction. To explicitly address this, we introduce a *semantic interaction path* that adaptively calibrates channel activations and enhances semantic-scale consistency across domains.

The semantic interaction operator employs two complementary projections—channel–height and channel–width—to capture cross-dimensional dependencies:

$$\mathcal{T}_{se}(X) = \tfrac{1}{2}\big(\mathcal{A}_{ch}(X^{(C,H)}) + \mathcal{A}_{cw}(X^{(C,W)})\big), \quad (6)$$

where $\mathcal{T}_{se}(X)$ denotes the aggregated semantic interaction feature. $\mathcal{A}_{ch}$ and $\mathcal{A}_{cw}$ denote channel–spatial attention operations defined in the spatial path. This formulation allows the network to jointly reason about semantic context along orthogonal spatial directions, yielding more discriminative and scale-invariant channel responses.

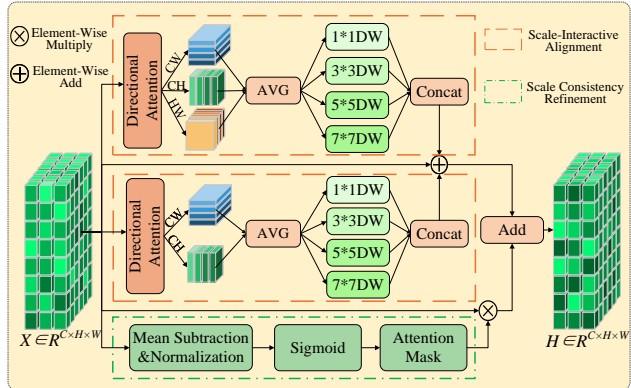

*Figure 3.* Details of the proposed Scale-Aware Domain Harmonization (SDH) module. SDH contains Scale-Interactive Alignment (SIA) and Scale Consistency Refinement (SCR). SIA uses spatial and semantic interaction paths to respectively harmonize geometric layout and channel-conditioned identity responses with directional attention and multi-scale depthwise convolutions. SCR further refines local activation consistency through mean-subtraction, normalization, and sigmoid masking, producing the final scale-harmonized feature $H$.

After attention-based modulation, a multi-scale convolutional refinement further integrates semantic information at multiple receptive levels:

$$F_{se} = \mathcal{M}_{se}(\mathcal{T}_{se}(X)), \quad (7)$$

where $\mathcal{F}_{se}$ denotes semantic path feature. $\mathcal{M}_{se}(\cdot)$ aggregates fine-grained and high-level semantics to enhance feature granularity.

Finally, the two interaction paths are adaptively fused with the original feature via a residual shortcut:

$$Y = \alpha F^{sp} + \beta F^{se} + X, \quad (8)$$

where $\alpha$ and $\beta$ are learnable balancing weights. This mechanism enables the model to jointly calibrate semantic and spatial scales, achieving consistent feature alignment across domains.

**Scale Consistency Refinement (SCR).** While SIA effectively harmonizes multi-scale representations, residual inconsistencies may still arise due to local distortions or background interference. To further stabilize the feature distribution, SCR performs a parameter-free refinement directly on the raw feature map $X \in \mathbb{R}^{C \times H \times W}$, aiming to enhance local activation consistency across spatial scales. Specifically, SCR estimates and normalizes spatial activation dispersion to measure local uncertainty:

$$\hat{Y} = \frac{(X - \mu(X))^2}{4\Big(\frac{1}{HW}\sum_{i,j}(X_{i,j} - \mu(X))^2 + \epsilon_s\Big)} + \eta, \quad (9)$$

where $\mu(X)$ denotes the spatial mean per channel, and $\epsilon_s$, $\eta$ are small constants controlling stability and bias. A sigmoid

gate then produces a saliency-aware modulation mask:

$$M = \sigma(\hat{Y}), \quad H^{\text{SCR}} = X \otimes M, \qquad (10)$$

where $\sigma(\cdot)$ is the sigmoid function and $\otimes$ denotes element-wise multiplication. Finally, the harmonized representation integrates semantic-scale calibration from SIA and local consistency refinement from SCR:

$$H = \alpha H^{\text{SIA}} + \beta H^{\text{SCR}} + \gamma X, \qquad (11)$$

where $\alpha, \beta, \gamma$ are learnable or empirically balanced coefficients. This collaborative design preserves structural fidelity while maintaining scale-consistent activation patterns across domains.

SIA and SCR jointly establish a closed-loop harmonization mechanism: SIA aligns semantic and structural representations across scales, while SCR reinforces them via saliency-guided refinement. Together, they preserve both semantic coherence and spatial uniformity, effectively mitigating scale discrepancies in cross-domain person search. Overall, SDH forms a hierarchical calibration framework that unifies cross-scale alignment and residual refinement, enhancing scale robustness and domain generalization across detection and re-identification branches.

## 2.3. Bidirectional Cluster Regularization (BCR)

To further enhance the robustness of pseudo-label learning, we propose a Bidirectional Cluster Regularization (BCR) module built upon the scale-consistent feature representations achieved by SDH. BCR adaptively regularizes both intra-cluster and inter-cluster structures through a split–merge correction, effectively reducing clustering noise and stabilizing self-training in the target domain. Specifically, it constructs split and merge matrices to regularize the original distance matrix, thereby refining pseudo labels in the target-domain feature space and improving the reliability of subsequent clustering and identity learning.

**Split Regularization (Intra-cluster refinement).** Given feature embeddings $f \in \mathbb{R}^{N \times d}$ and pseudo labels $\hat{y}$, BCR first examines the compactness of each cluster $C_m$. the mean intra-cluster affinity is computed as:

$$a_m = \frac{1}{|\mathcal{C}_m|} \sum_{x_i \in \mathcal{C}_m} \frac{1}{|\mathcal{C}_m| - 1} \sum_{x_j \in \mathcal{C}_m, j \neq i} \|f_i - f_j\|_2, \quad (12)$$

where $f_i$ denotes the feature vector of sample $x_i$, and $a_m$ denotes the mean intra-cluster affinity. A large $a_m$ indicates that $C_m$ is spatially loose and should be further split. When a cluster is identified as internally inconsistent, we subdivide it by spectral sub-clustering and apply a positive penalty to pairs across different sub-clusters. The split correction is summarized as:

$$\mathbf{M}_s[p, q] = \min\left(\frac{\varepsilon}{\lambda} \lambda^{a_m}, \text{clip}_{splitmax}\right), \qquad (13)$$

---

**Algorithm 1** Bidirectional Cluster Regularization (BCR)

1: **Input:** Target features $f \in \mathbb{R}^{N \times d}$, pseudo labels $\hat{y}$, base distance $\mathbf{D}_{base}$, scaling parameters $\varepsilon, \lambda$
2: **Output:** Corrected distance matrix $\mathbf{D}_{corr}$
3: **Split Regularization**
4: **for** each cluster $\mathcal{C}_m$ in $\hat{y}$ **do**
5:      Compute mean intra-cluster distance $a_m$
6:      **if** $a_m > \tau_{split}$ **then**
7:          Split $\mathcal{C}_m$ into $K$ sub-clusters via spectral clustering
8:          For $(x_p, x_q)$ from different sub-clusters: $\mathbf{M}_s[p, q] = \min(\frac{\varepsilon}{\lambda} \lambda^{a_m}, \text{clip}_{splitmax})$
9:      **end if**
10: **end for**
11: **Merge Regularization**
12: **for** each cluster $\mathcal{C}_m$ **do**
13:      Find nearest cluster $\mathcal{C}_p$ and compute mean inter-cluster distance $b_m$
14:      Compute vote ratio $\rho$ between $(\mathcal{C}_m, \mathcal{C}_p)$
15:      **if** $\rho > \tau_{vote}$ and $\bar{d}_{ij} < \tau_{merge}$ **then**
16:          For $(x_p \in \mathcal{C}_m, x_q \in \mathcal{C}_p)$: $\mathbf{M}_m[p, q] = \max(-\frac{\varepsilon}{\lambda} \lambda^{(1-b_m)}, -\text{clip}_{mergemax})$
17:      **end if**
18: **end for**
19: **Distance Correction**
20: $\mathbf{D}_{corr} = \text{sym}(\mathbf{D}_{base} + \mathbf{M}_s + \mathbf{M}_m)$
21: Set diagonal to zero and clip negatives: $\mathbf{D}_{corr} \leftarrow \max(\mathbf{D}_{corr}, 0)$
22: **return** $\mathbf{D}_{corr}$

---

where $\varepsilon$ and $\lambda$ are scaling parameters controlling the split strength, and $\text{clip}_{splitmax}$ limits the maximum penalty to prevent over-splitting. Larger $a_m$ values yield stronger split penalties, encouraging the separation of structurally incoherent clusters. Specifically, Eq. (13) $\mathbf{M}_s[p, q]$ is applied only when $x_p$ and $x_q$ belong to the same cluster $\mathcal{C}_m$ but are assigned to different sub-clusters; otherwise, $\mathbf{M}_s[p, q] = 0$. We only attempt splitting for clusters with sufficient size to avoid unreliable splits and with mean intra-cluster distance above a designated threshold. The threshold and sub-cluster count are hyperparameters (denoted $\tau_{\text{split}}$ and $K$ in the implementation). Intuitively, $\mathbf{M}_s$ increases pairwise distances between confidently separated sub-groups to discourage incorrect aggregation in subsequent clustering.

**Merge Regularization (Inter-cluster refinement).** To prevent over-fragmentation, we further compute the mean inter-cluster distance between $\mathcal{C}_m$ and its nearest neighbor $\mathcal{C}_p$ as:

$$b_m = \frac{1}{|\mathcal{C}_m|} \sum_{x_i \in \mathcal{C}_m} \min_{\mathcal{C}_p \neq \mathcal{C}_m} \frac{1}{|\mathcal{C}_p|} \sum_{x_j \in \mathcal{C}_p} \|f_i - f_j\|_2, \quad (14)$$

where $b_m$ denotes the mean inter-cluster affinity. Smaller $b_m$ values imply strong semantic affinity between clusters,

suggesting that merging should be encouraged. The corresponding merge correction matrix $\mathbf{M}_m$ is formulated as:

$$\mathbf{M}_m[p,q] = \max\left( -\frac{\varepsilon}{\lambda}\lambda^{(1-b_m)}, -\text{clip}_{mergemax} \right), \quad (15)$$

where $\varepsilon$ and $\lambda$ are scaling coefficients controlling the magnitude of merge correction, $\text{clip}_{mergemax}$ limits the maximum negative correction to avoid excessive merging, and $b_m$ denotes the average inter-cluster distance between cluster $\mathcal{C}_m$ and its nearest neighbor $\mathcal{C}_p$. Specifically, Eq. (15) $\mathbf{M}_m[p,q]$ is applied only when $x_p$ and $x_q$ belong to two distinct clusters $\mathcal{C}_m$ and $\mathcal{C}_p$ that satisfy the merge criteria; otherwise, $\mathbf{M}_m[p,q] = 0$. We only attempt merging for cluster pairs that exhibit high mutual affinity and consistent nearest-neighbor voting patterns. In particular, a merge is triggered when the inter-cluster distance $\bar{d}_{mp}$ falls below the merging threshold $\tau_{\text{merge}}$ and the vote ratio between $\mathcal{C}_m$ and $\mathcal{C}_p$ exceeds a reliability ratio $\rho$. Both $\tau_{\text{merge}}$ and $\rho$ are hyperparameters controlling merge sensitivity. Intuitively, $\mathbf{M}_m$ reduces the pairwise distances between mutually consistent clusters, encouraging them to be grouped together in subsequent clustering iterations.

**Distance Correction.** The split and merge corrections are combined with the base distance matrix to produce a corrected, symmetric distance used for reclustering:

$$\mathbf{D}_{\text{corr}} = \text{sym}\big(\mathbf{D}_{\text{base}} + \mathbf{M}_s + \mathbf{M}_m\big), \quad (16)$$

where $\mathbf{D}_{\text{corr}} \in \mathbb{R}^{N \times N}$ denotes the corrected distance matrix, and $\mathbf{D}_{\text{base}}$ represents the original pairwise distance. $\mathbf{M}_s$ and $\mathbf{M}_m$ are the split and merge correction matrices defined in Eq. (13) and Eq. (15), respectively. $\text{sym}(\cdot)$ denotes symmetric averaging, i.e., $\text{sym}(\mathbf{A}) = (\mathbf{A} + \mathbf{A}^\top)/2$. The diagonal entries of $\mathbf{D}_{\text{corr}}$ are set to zero to ensure a valid self-distance constraint.

By balancing the split and merge forces, BCR adaptively regularizes the evolving pseudo-label distribution: $\mathbf{M}_s$ discourages false positives within loose clusters, while $\mathbf{M}_m$ recovers missed associations between semantically consistent clusters.

# 3. Experiments

## 3.1. Experimental Settings

**Datasets.** We evaluate the proposed SCALE framework on two widely used person search benchmarks, CUHK-SYSU (Xiao et al., 2017) and PRW (Zhong et al., 2017). The CUHK-SYSU dataset is a large-scale person search benchmark containing 18,184 images collected from handheld cameras, movies, and TV shows, which introduces substantial scene diversity. It encompasses 8,432 unique person identities and 96,143 annotated bounding boxes. The dataset

is divided into a training set with 5,532 identities and 11,206 images, and a test set with 2,900 query persons and 6,978 gallery images. For each query, the dataset defines a gallery size ranging from 50 to 4,000, with a default size of 100 images. The PRW dataset consists of video frames captured by six fixed surveillance cameras on a university campus. It contains 11,816 scene images with 932 distinct identities and 43,110 annotated bounding boxes. The training set includes 483 identities and 5,704 images, while the test set comprises 2,057 query persons and 6,112 gallery images. For evaluation, all images in the test set except the query are used as the gallery.

**Evaluation Metric.** Following previous person search studies, we adopt mean Average Precision (mAP) and top-1 accuracy (top-1) as the evaluation metrics to comprehensively assess the performance of SCALE. Specifically, mAP measures the retrieval precision across different recall levels, providing an overall evaluation of ranking quality, while top-1 reflects the identification accuracy by computing the proportion of queries whose correct match appears at the top of the ranked list. Both metrics are calculated under the standard gallery settings defined for each dataset.

**Implementation Details.** We implemented our SCALE framework using PyTorch (Paszke et al., 2019) and trained it on an NVIDIA A800 GPU with a batch size of 4. We adopted the Stochastic Gradient Descent (SGD) optimizer with a learning rate of 0.0024, which is linearly warmed up during the first epoch. We utilize ResNet-50 (He et al., 2016) pre-trained on ImageNet-1k (Deng et al., 2009) as the default backbone network. During training, input images are resized to 1500×900, and random horizontal flipping is applied for data augmentation. To prevent overfitting, we adjust the total number of training epochs based on the target-domain dataset. Specifically, when PRW serves as the target domain, SCALE is first pre-trained on the source domain CUHK-SYSU for 8 epochs, followed by joint training for 13 epochs. Conversely, when CUHK-SYSU is the target domain, we pre-train on PRW for 3 epochs before performing 15 epochs of joint training.

## 3.2. Comparison with State-of-the-Art Methods

We compare SCALE with recent fully supervised, weakly supervised, and unsupervised person search methods on the CUHK-SYSU and PRW datasets, as summarized in Table 1. Among unsupervised approaches, SCALE achieves the best overall performance, reaching 41.7% mAP / 82.4% top-1 on PRW and 82.3% mAP / 84.0% top-1 on CUHK-SYSU, surpassing the previous best-performing DSCA (Qi et al., 2025) by +1.8 mAP and +2.1 mAP on the two benchmarks, respectively. Compared with fully supervised and weakly supervised approaches, SCALE remarkably narrows the performance gap, despite not using any target-domain ground

| Category | Method | Venue | Backbone | PRW | | CUHK-SYSU | |
|---|---|---|---|---|---|---|---|
| | | | | mAP | top-1 | mAP | top-1 |
| Fully-Supervised | OIM (Xiao et al., 2017) | CVPR2017 | ResNet-50 | 21.3 | 49.4 | 75.5 | 78.7 |
| | MGTS (Chen et al., 2018) | ECCV2018 | VGG-16 | 32.6 | 72.1 | 83.0 | 83.7 |
| | RDLR (Han et al., 2019) | ICCV2019 | ResNet-50 | 42.9 | 70.2 | 93.0 | 94.2 |
| | NAE+ (Chen et al., 2020) | CVPR2020 | ResNet-50 | 44.0 | 81.1 | 92.1 | 92.9 |
| | AlignPS+ (Yan et al., 2021) | CVPR 2021 | ResNet-50 | 46.1 | 82.1 | 94.0 | 94.5 |
| | SeqNet (Li & Miao, 2021) | AAAI2021 | ResNet-50 | 46.7 | 83.4 | 93.8 | 94.6 |
| | PSTR (Cao et al., 2022) | CVPR2022 | PVTv2-B2 | 56.5 | 89.7 | 95.2 | 96.2 |
| | SeqNeXt (Jaffe & Zakhor, 2023) | WACV2023 | ConvNeXt-B | 57.6 | 89.5 | 96.1 | 96.5 |
| | SEAS (Jiang et al., 2024) | IJCAI2024 | ConvNeXt-B | 60.5 | 89.5 | 97.1 | 97.8 |
| Weakly-Supervised | CGPS (Yan et al., 2022) | AAAI2022 | ResNet-50 | 16.2 | 68.0 | 80.0 | 82.3 |
| | R-SiamNet (Han et al., 2021) | ICCV2021 | ResNet-50 | 21.4 | 75.2 | 86.0 | 87.1 |
| | SSL (Wang et al., 2023) | ICCV2023 | ResNet-50 | 33.9 | 82.7 | 87.6 | 89.0 |
| | DICL (Wang et al., 2024) | PR2024 | ResNet-50 | 35.5 | 80.9 | 87.4 | 88.8 |
| Unsupervised | DAPS (Li et al., 2022) | ECCV2022 | ResNet-50 | 34.7 | 80.6 | 77.6 | 79.6 |
| | FOUS (Cui et al., 2024) | IJCAI2024 | ResNet-50 | 35.4 | 80.8 | 78.7 | 80.5 |
| | DDAM (Almansoori et al., 2024) | WACV2024 | ResNet-50 | 36.7 | 81.2 | 79.5 | 81.3 |
| | MoS (Kim et al., 2025) | CVPR2025 | ResNet-50 | 37.1 | _81.9_ | 80.1 | 81.5 |
| | DSCA (Qi et al., 2025) | AAAI2025 | ResNet-50 | _39.9_ | 81.6 | _80.2_ | _81.7_ |
| | **SCALE (Ours)** | | ResNet-50 | **41.7** | **82.4** | **82.3** | **84.0** |

*Table 1.* Comparison of mAP(%) and top-1 accuracy(%) with fully supervised, weakly supervised, and domain adaptation methods on CUHK-SYSU and PRW test set. The best and second-best results among domain adaptation methods are in bold and underlined.

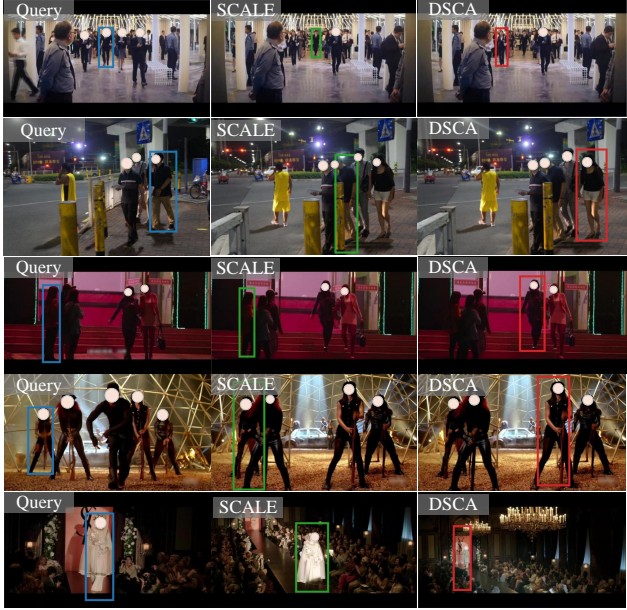

*Figure 4.* Qualitative comparison of SCALE with DSCA on the CUHK-SYSU test set. The blue bounding boxes denote the queries, while the green and red bounding boxes denote correct and incorrect matches, respectively.

truth identity labels.

**Qualitative Comparison.** To further validate the effectiveness of our method, we present a qualitative comparison between SCALE and DSCA on the CUHK-SYSU test set, as shown in Figure 4. The visualization shows that SCALE produces more accurate detections and retrievals under chal-

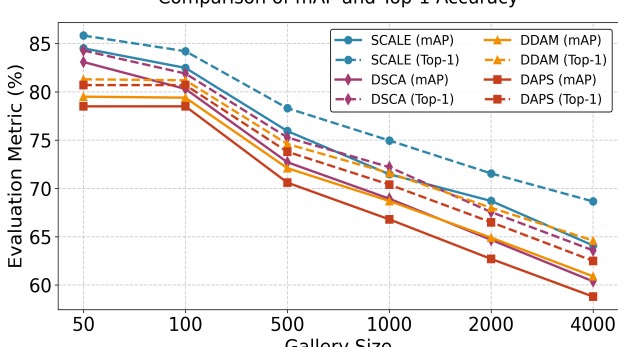

*Figure 5.* Comparison of mAP and top-1 accuracy on CUHK-SYSU across various gallery sizes, where solid lines denote mAP and dashed lines denote top-1 accuracy.

lenging conditions such as occlusion, scale variation, and domain shift. Benefiting from the SDH module, the model maintains consistent attention across pedestrians of different scales. Meanwhile, the BCR module effectively suppresses pseudo-label noise, thus leading to cleaner and more reliable retrieval results.

**Scalability to Gallery Size.** Figure 5 further investigates the impact of gallery size on performance. As the gallery size increases from 50 to 4000, SCALE consistently outperforms previous unsupervised methods, maintaining a slower degradation rate in both mAP and top-1 accuracy. This robustness under large-scale retrieval settings highlights the discriminative and domain-consistent representations learned by the framework.

Overall, the results in Table 1, Figure 4 and Figure 5 collectively demonstrate that the SCALE framework effectively mitigates both feature-scale inconsistencies and pseudo-label noise in the unsupervised person search setting, achieving robust, label-free domain adaptation with strong generalization to challenging real-world scenarios.

## 3.3. Ablation Study

To validate the effectiveness of each proposed component, we conduct a series of ablation studies on the PRW and CUHK-SYSU datasets. Table 2 reports the results of ablating the key components of our framework, including the Scale-Interactive Alignment (SIA), Scale Consistency Refinement (SCR), and Bidirectional Cluster Regularization (BCR). Removing any component leads to a noticeable performance drop, verifying that all modules contribute complementarily to the overall system. Specifically, eliminating SIA results in a clear decrease in both mAP and top-1 accuracy (–0.6% and –0.5% on PRW), highlighting the importance of scale-invariant domain alignment. Removing SCR weakens feature-level semantic consistency, leading to unstable domain harmonization and degraded accuracy. Meanwhile, removing BCR harms the stability of pseudo-label refinement, reducing the reliability of clustering-based self-training. When multiple components are simultaneously removed, performance further declines, demonstrating that SIA, SCR, and BCR work synergistically to enhance cross-domain robustness and representation quality.

| Components | | | PRW | | CUHK-SYSU | |
|---|---|---|---|---|---|---|
| SIA | SCR | BCR | mAP | top-1 | mAP | top-1 |
| | | | 41.7 | 82.4 | 82.3 | 84.0 |
| ✗ | ✗ | ✗ | 39.9 | 81.6 | 80.2 | 81.7 |
| ✗ | ✗ | | 40.8 | 81.7 | 81.2 | 83.3 |
| ✗ | | ✗ | 40.6 | 81.5 | 81.3 | 83.3 |
| | ✗ | ✗ | 40.9 | 81.7 | 81.1 | 82.7 |
| ✗ | | | 41.1 | 81.9 | 81.4 | 83.0 |
| | ✗ | | 41.4 | 82.2 | 81.7 | 83.7 |
| | | ✗ | 41.3 | 82.0 | 81.7 | 83.5 |

*Table 2.* Component ablation study on PRW and CUHK-SYSU datasets, including SIA, SCR, and BCR. ✗ indicates the component is ablated.

Furthermore, we investigate the impact of the number of sub-clusters $n$ used in the Split branch, which determines the granularity of intra-cluster refinement. As summarized in Table 3, performance first improves as $n$ increases from 1 to 2, indicating that moderate subdivision helps better capture local semantic structures within noisy clusters. However, an excessive number of sub-clusters (e.g., $n = 3$) causes over-segmentation and unstable optimization, leading to performance degradation. Therefore, we set $n = 2$ as the optimal configuration for all experiments.

| $n$ | PRW | | CUHK-SYSU | |
|---|---|---|---|---|
| | mAP | top-1 | mAP | top-1 |
| 1 | 40.3 | 81.5 | 80.4 | 82.0 |
| 2 | 40.8 | 81.7 | 81.2 | 83.3 |
| 3 | 39.6 | 80.7 | 79.8 | 80.5 |

*Table 3.* Performance on the benchmark CUHK-SYSU and PRW datasets with different numbers of sub-clusters in the Split sub-branch of BCR.

| Components | | PRW | | CUHK-SYSU | |
|---|---|---|---|---|---|
| Split | Merge | mAP | top-1 | mAP | top-1 |
| | | 40.8 | 81.7 | 81.2 | 83.3 |
| ✗ | | 40.3 | 81.5 | 80.4 | 82.0 |
| | ✗ | 40.5 | 82.0 | 80.7 | 82.5 |
| ✗ | ✗ | 39.9 | 81.6 | 80.2 | 81.7 |

*Table 4.* Component ablation study on PRW and CUHK-SYSU datasets. Split and Merge are key components of BCR, with ✗ indicating the component is ablated.

To further analyze the contribution of each component within the proposed Bidirectional Cluster Regularization (BCR), we conduct additional ablation studies focusing on its Split and Merge sub-branches. As shown in Table 4, removing either sub-branch leads to a noticeable performance drop on both PRW and CUHK-SYSU datasets, demonstrating that Split and Merge jointly enhance the reliability of pseudo-label refinement. Specifically, disabling the Split operation results in less compact clustering, while removing Merge weakens the correction of inter-cluster inconsistencies, confirming that both are essential for stable self-training.

## 3.4. Further Analysis

Beyond the standard comparison and component ablations, we further analyze SCALE from three perspectives: whether cross-domain scale inconsistency indeed exists in real person search datasets, whether the proposed method is especially beneficial under scale-mismatched cases, and whether the improvement comes from the proposed scale-aware design rather than generic feature refinement or increased model capacity.

**Quantifying cross-domain scale inconsistency.** To directly verify the motivation of SCALE, we first analyze the bounding-box area distributions of pedestrians in CUHK-SYSU and PRW. The statistics are used only for analysis and are never involved in model training. As shown in Table 5, the two datasets exhibit a clear scale distribution shift. In particular, CUHK-SYSU contains a much larger proportion of very small pedestrians in the 0–5k area range, while PRW has more samples concentrated in the 5k–10k and larger-scale ranges. This indicates that models transferred

| Area Range | CUHK-SYSU | PRW |
|---|---|---|
| 0–1k | 12.2 | 0.0 |
| 1k–5k | 33.0 | 20.6 |
| 5k–10k | 15.7 | 31.9 |
| 10k–30k | 24.2 | 27.9 |
| >30k | 14.8 | 19.6 |

*Table 5.* Bounding-box scale distribution across CUHK-SYSU and PRW. Values are dataset proportions (%).

between the two datasets encounter substantially different RoI spatial supports. Such a shift is especially harmful for person search, since scale variation affects both proposal localization in the detection branch and identity representation in the ReID branch. Unlike conventional ReID or classification tasks that usually operate on pre-cropped or scale-normalized instances, person search extracts identity features from detector-generated RoIs. Thus, inaccurate localization caused by small-scale observations can directly corrupt the spatial support used for ReID feature learning.

**Effectiveness under scale mismatch.** We then examine whether the gain of SCALE becomes more evident when scale inconsistency is emphasized. Under the scale-mismatched setting, where target-domain pedestrians are evaluated in the most scale-discrepant range, i.e., the area < 10k subset, SCALE improves the baseline from 53.53% / 59.79% to 57.49% / 64.14% in terms of mAP/top-1, yielding absolute gains of +3.96% mAP and +4.35% top-1. These gains are more pronounced than those under the standard benchmark setting, indicating that SCALE is particularly effective when target samples suffer from insufficient spatial support and noisy identity cues caused by scale variation.

**Comparison with simpler scale-oriented variants.** To further verify that the improvement is not merely brought by stronger multi-scale processing or additional attention modules, we compare SCALE with several stronger controlled variants in Table 6, including multi-scale training/testing, explicit size normalization, FPN-style modification, and inserting similar refinement modules without the proposed scale-aware harmonization design. Although these variants provide limited improvements, they consistently underperform SCALE. For example, the FPN-style variant reaches 81.1% mAP and 82.7% top-1, while SCALE achieves 82.3% mAP and 84.0% top-1. This comparison shows that simply enlarging the receptive field or adding generic feature refinement is insufficient, and the proposed spatial-semantic scale harmonization is critical for robust domain adaptation in person search.

**Complementarity of spatial and semantic paths.** We also evaluate whether the two paths in SDH capture complementary information. As shown in Table 6, replacing SDH with double spatial paths, double semantic paths, or shared-parameter paths all leads to lower performance than the full SDH. This confirms that the spatial path and semantic path are not interchangeable. The spatial path mainly stabilizes

| Method | mAP | top-1 |
|---|---|---|
| *Scale-oriented and parameter-matched variants* | | |
| Multi-scale training/testing | 80.6 | 81.9 |
| Explicit size normalization | 80.4 | 81.9 |
| FPN-style modification | 81.1 | 82.7 |
| Similar refinement modules | 80.7 | 82.6 |
| SCALE | **82.3** | **84.0** |
| *Spatial-semantic path variants* | | |
| Double spatial path | 81.0 | 83.1 |
| Double semantic path | 80.7 | 82.5 |
| Shared parameters | 80.6 | 81.9 |
| Full SDH | **82.3** | **84.0** |

*Table 6.* Controlled comparisons on CUHK-SYSU. The upper block compares simpler scale-oriented variants, and the lower block analyzes the spatial-semantic paths in SDH.

geometric layout under varying pedestrian scales, while the semantic path calibrates channel-conditioned identity responses. Their combination is important for person search, where inaccurate localization and corrupted identity features can mutually amplify each other under cross-domain scale variation.

**Computational cost.** SCALE introduces only moderate overhead compared with the baseline, increasing the parameters from 56.19M to 58.64M and the computation from 414.09 GMac to 459.90 GMac, corresponding to +4.4% parameters and +11.1% computation. Since the overall network is still dominated by backbone feature extraction, the additional cost of SDH is acceptable. BCR reuses the same pairwise distance matrix as the baseline clustering pipeline and only performs localized split–merge corrections, so the overall clustering complexity remains $O(N^2)$.

## 4. Conclusion

In this paper, we proposed SCALE, a novel framework for unsupervised domain-adaptive person search that effectively bridges cross-domain scale discrepancies between source and target domains. To address the scale inconsistency naturally arising from pedestrian distances, camera viewpoints, and environmental conditions across scenes, we introduced the Scale-aware Domain Harmonization (SDH) module, which adaptively aligns cross-domain features through hierarchical scale interaction and consistency refinement. Moreover, our Bidirectional Cluster Regularization (BCR) explicitly enhances the structural reliability of clustering-based pseudo labels by enforcing split–merge consistency, ensuring both finer separation of loose clusters and robust aggregation of semantically aligned ones. Extensive experiments on multiple benchmarks demonstrate that our method substantially improves cross-domain generalization and achieves state-of-the-art performance.

## Acknowledgments

This work was supported in part by National Natural Science Foundation of China Grant 62576067, 62501226, the National Key Research and Development Program of China Grant 2024YFB4710800, Liaoning Provincial Natural Science Foundation Grant 2025-YQ-01, 2024 MS-012 and 2025-BS-0233, Natural Science Foundation of Hebei Province F2025201037, Dalian Science and Technology Talent Innovation Support Plan Grant 2024RY010, Ministry of Industry and Information Technology of the People's Republic of China under Grant No. 18Q-25-06, the China Postdoctoral Science Foundation, China (Grant Number: 2024M760315 and 2025T180437).

## Impact Statement

This paper presents research aimed at advancing unsupervised representation learning and cross-domain generalization, with person search serving as a challenging benchmark task. By reducing the reliance on manual annotations, the proposed method contributes to more scalable and adaptable learning paradigms in machine learning.

Like many visual recognition and retrieval technologies, person search methods may have broader societal implications depending on how they are deployed in real-world systems. Potential concerns include privacy, data bias, and misuse in large-scale surveillance scenarios. We note that this work does not involve identity recognition, personal identification, or deployment in real-world applications, and all experiments are conducted on publicly available academic benchmarks.

We encourage that future applications of this research be developed and used in accordance with relevant legal, ethical, and societal guidelines.

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
