# OpenReview forum: "Scale-Aware Domain Harmonization for Domain Adaptation Person Search"
_ICML.cc/2026/Conference — ICML 2026 regular_

### Official Review · Reviewer_XZPW · 2026-03-06

**Soundness:** 3
**Presentation:** 2
**Significance:** 1
**Originality:** 2
**Overall Recommendation:** 2
**Confidence:** 4

**Summary:**

This paper aims at unsupervised domain adaptation (UDA) for person search and attributes the cross-domain performance degradation mainly to scale inconsistency across datasets. To address this, the authors propose SCALE, a framework that introduces a Scale-aware Domain Harmonization (SDH) module to enhance feature alignment across domains, along with a Bidirectional Clustering Refinement (BCR) strategy to improve pseudo-label quality during training.

**Compliance With Llm Reviewing Policy:**

Affirmed.

**Final Justification:**

The paper addresses UDA for person search and identifies scale inconsistency as a key factor. While the rebuttal has satisfactorily addressed my concerns regarding scale inconsistency, my main concern about task relevance remains.

Specifically, the proposed method mainly focuses on general UDA challenges such as feature alignment and pseudo-label refinement, without clearly demonstrating why it is particularly tailored to person search. As a result, the method appears to be a generic UDA solution that could be applied beyond this task, yet the paper does not provide sufficient empirical justification.

Therefore, I maintain my reject recommendation.

**Key Questions For Authors:**

Please refer to the concerns raised in the **Weaknesses** section. In particular, the authors are encouraged to clarify how the proposed method specifically addresses the scale inconsistency issue claimed in the paper.

**Limitations:**

Yes

**Strengths And Weaknesses:**

**Strengths**

The paper investigates unsupervised domain adaptation for person search, which is a practical and challenging problem since models trained on one dataset often suffer from significant performance drops when deployed in new environments.

---


**Weaknesses**

1. **Presentation**

   The introduction section contains an overly detailed discussion of many UDA-related works, and the presentation is somewhat scattered. Such detailed literature descriptions would be more appropriate in the Related Work section. In the introduction, the discussion should be more concise and structured, focusing on clearly explaining **why existing methods are insufficient to address the challenges of UDA**, which would help better motivate the proposed approach.

2. **Task Relevance**

   The connection between the proposed method and the person search task is not entirely clear. From the description in the paper, most of the technical components focus on addressing unsupervised domain adaptation (UDA) issues, such as cross-domain feature alignment and pseudo-label refinement. However, these problems are largely generic UDA challenges, rather than issues specific to the person search task itself. It would be helpful for the authors to clarify why the proposed design is particularly suitable for person search, instead of being a general UDA solution..

3. **Method Design**

   While the paper attributes the cross-domain gap to scale inconsistency, it is not entirely clear how the proposed SDH module explicitly models or aligns scale differences. From the method description, SDH mainly performs feature interaction and attention-based refinement along spatial and channel dimensions. However, these operations appear to function as general feature recalibration mechanisms, rather than explicitly addressing scale variations. As a result, the method may work, but the explanation is not convincing.

4. **Experimental Evaluation**

   The experimental validation raises some concerns:

   (1) While the paper introduces the BCR module for clustering refinement, the experiments only compare internal variants of BCR and do not evaluate it against existing pseudo-label or clustering refinement strategies commonly used in UDA settings.

   (2) Although the paper attributes the cross-domain gap mainly to scale inconsistency, the experiments do not provide quantitative analysis of scale differences across datasets, nor do they conduct scale-specific evaluation. As a result, the claim that the proposed method effectively addresses scale inconsistency is not sufficiently supported by the experimental evidence.

---

> ### Author Rebuttal · Authors · 2026-03-31
>
> We thank the reviewer for the thoughtful and constructive feedback. We address the key concerns as follows.
>
> **Q1:** why existing methods are insufficient to address the challenges of UDA
>
> **A1:** The introduction is structured to progressively build the problem setting, from general UDA to the specific challenges in person search. Prior works are discussed to highlight that existing methods mainly focus on appearance alignment, which motivates the identification of a less-explored issue—cross-domain scale inconsistency. This progression is illustrated in Fig. 1, where the motivation of scale harmonization is derived from limitations of prior approaches. Due to space constraints of the conference, part of the related work is included in the introduction.
>
> **Q2:** Task relevance to person search.
>
> **A2:** Although the proposed method is developed under the UDA setting, it is specifically motivated by the characteristics of person search. Unlike conventional UDA tasks, which typically consider only feature alignment for a single recognition objective, person search integrates two tightly coupled sub-tasks: person detection and person re-identification. Therefore, scale variation introduces a dual challenge, degrading both detection quality through inaccurate bounding boxes and ReID performance through inconsistent identity representations. As a result, cross-domain scale inconsistency introduces coupled degradation in localization and retrieval, making it a task-specific challenge. Our design explicitly targets this issue by jointly modeling spatial structure and semantic representation, which is particularly important for person search pipelines. To demonstrate the effectiveness of our spatial structure and semantic representation, we further verify through experiments that removing either spatial or semantic interaction degrades the performance, as shown in the table below.
>
> | method               | mAP  | top1 |
> | -------------------- | ---- | ---- |
> | double spatial path  | 81.0 | 83.1 |
> | double semantic path | 80.7 | 82.5 |
> | share parameters     | 80.6 | 81.9 |
> | SDH                  | 82.3 | 84.0 |
>
> **Q3:** Method Design: Connection between SDH and scale modeling.
>
> **A3:** SDH is motivated by how scale inconsistency alters feature representations across domains. Specifically, variations in person scale lead to mismatched spatial configurations as well as shifts in semantic granularity. Based on this, SDH decomposes the adaptation process into two coordinated operations: spatial interaction for modeling scale-sensitive geometric patterns, and channel-conditioned interaction for adjusting feature responses across semantic levels. Therefore, although implemented with attention-style operations, SDH is not intended as a generic refinement block; rather, its design is tailored to address the structural and semantic misalignment induced by scale variation. We further verify that SDH is not a simple feature refinement module by comparing it with attention mechanisms of similar parameter count and simple multi-scale modules, as shown in the table below.
>
> | method                                   | mAP  | top1 |
> | ---------------------------------------- | ---- | ---- |
> |  multi-scale                          | 80.6 | 81.9 |
> |  FPN-style modifications              | 81.1 | 82.7 |
> |  inserting similar refinement modules | 80.7 | 82.6 |
> | ours                                     | 82.3 | 84.0 |
>
> **Q4:** Experimental Evaluation. Lacks (i) comparison with existing clustering methods and (ii) direct evidence for the scale inconsistency claim.
>
> **A4:** The current manuscript does include comparisons against representative UDA methods that rely on pseudo-label generation and clustering-based refinement, such as DAPS, DDAM, and DSCA, as reported in Table 1. These methods typically construct a Jaccard distance matrix and perform DBSCAN clustering based on this distance for pseudo-label generation. Our original presentation, however, did not make this connection sufficiently explicit, which may have caused confusion. We will revise the paper to clearly state that these methods serve as relevant baselines for evaluating BCR under the common clustering-based UDA setting.
>
> For the scale inconsistency claim, we design a controlled evaluation protocol to simulate cross-domain scale mismatch. Specifically, we perturb the target bounding boxes by systematically reducing their spatial extent, which mimics scenarios where the target domain contains smaller-scale people compared to the source domain. Under this setting, we observe that the baseline performance drops significantly, while our method remains more robust. This result indicates that the improvement is not merely a general performance gain, but is particularly pronounced when scale inconsistency is present.
> | **method** | **mAP** | **top1** |
> | ---------- | ------- | -------- |
> | baseline   | 53.53   | 59.79    |
> | SCALE      | 57.49   | 64.14    |

---

> > ### Author Rebuttal · Reviewer_XZPW · 2026-04-04
> >
> > Thank you for your detailed rebuttal. I appreciate the clarifications and the effort to address my concerns. Some of my questions have been partially addressed; however, certain concerns remain:
> >
> > 1. **Task relevance to person search**: Your response emphasizes the differences between person search and conventional UDA tasks. My concern, however, is about the **direct connection between the proposed method and person search**. The described spatial structure and semantic representation mechanisms also exist in general UDA tasks, suggesting that your method could potentially be applied beyond person search. It remains unclear what aspects make it specifically tailored to this task.
> > 2. **Scale inconsistency**: While you discuss a controlled evaluation with perturbed bounding boxes, you **do not provide quantitative results of scale differences across the actual datasets**. Without this, it is difficult to assess whether scale inconsistency truly impacts performance and to what extent your method mitigates it.
> >
> > Overall, the rebuttal clarifies some points but these issues remain open.

---

> > > ### Author Response · Authors · 2026-04-05
> > >
> > > **Task relevance to person search.**
> > > The proposed method is not defined by the use of spatial or semantic operations themselves, but by the specific failure mode it targets in person search. Unlike conventional UDA tasks that operate on pre-cropped and scale-normalized inputs, person search performs detection and re-identification jointly on unconstrained images, where feature extraction is directly conditioned on detection results.
> > >
> > > Under this setting, scale variation introduces a coupled error propagation: inaccurate bounding boxes at small scales lead to spatial misalignment (e.g., missing body parts or distorted layouts), which directly corrupts the spatial support for feature extraction. At the same time, small-scale observations inherently contain incomplete and noisy identity cues, leading to semantic corruption in feature representation, where unreliable local regions dominate the aggregation process.
> > >
> > > SDH is designed to explicitly mitigate this error propagation at its two sources. First, the spatial interaction path targets the misalignment introduced by inaccurate bounding boxes. By aggregating information across spatial projections and multiple receptive fields, it stabilizes geometric structure under varying scales and reduces sensitivity to imperfect spatial support.
> > > Second, the semantic interaction path addresses the corruption of identity features caused by incomplete observations at small scales. It constrains how features are aggregated under different spatial supports, preventing unreliable local regions from dominating the final representation.
> > >
> > > By jointly handling these two failure modes, SDH reduces the cross-stage inconsistency caused by scale variation, rather than acting as a generic feature refinement module. Empirically, removing either path leads to consistent performance degradation, confirming that both are necessary to resolve this task-specific coupling.
> > >
> > >
> > >
> > > **Scale inconsistency.**
> > >  We agree that directly quantifying scale differences across datasets is important. We first analyze the bounding box scale distributions in source and target domains, as shown in the table below, and observe a clear distribution shift, particularly in the **small-scale regime (area <10k)**.
> > >
> > > Based on this observation, we focus on the most discrepant range (<10k) and conduct controlled experiments under this setting. We find that the performance gap between the baseline and our method is significantly larger in this regime, indicating that small-scale instances are more affected by cross-domain scale mismatch.
> > >
> > > This provides direct evidence that the proposed method is particularly effective in mitigating this issue where the mismatch is most severe. These analyses will be included in the final version.
> > >
> > > | Area Range | CUHK-SYSU Dataset Proportion | PRW Dataset Proportion |
> > > | ---------- | ---------------------------- | ---------------------- |
> > > | 0–1k       | 12.2%                        | 0.0%                   |
> > > | 1k–5k      | 33.0%                        | 20.6%                  |
> > > | 5k–10k     | 15.7%                        | 31.9%                  |
> > > | 10k–30k    | 24.2%                        | 27.9%                  |
> > > | >30k       | 14.8%                        | 19.6%                  |

---

### Official Review · Reviewer_6iLh · 2026-03-10

**Soundness:** 3
**Presentation:** 3
**Significance:** 3
**Originality:** 4
**Overall Recommendation:** 5
**Confidence:** 5

**Summary:**

This paper proposes a SCALE framework to address the domain scale inconsistency issue in UDA person search. The framework comprises a Scale-aware Domain Harmonization (SDH) module for cross-domain scale alignment, as well as a Bidirectional Clustering Regularization (BCR) strategy that optimizes pseudo-labels via split-merge clustering calibration. Extensive experiments on the CUHK-SYSU and PRW datasets demonstrate that the proposed method achieves SOTA performance.

**Compliance With Llm Reviewing Policy:**

Affirmed.

**Final Justification:**

After reviewing all the comments and rebuttals, I think all the concerns have been well stressed.

**Key Questions For Authors:**

1.The proposed BCR module introduces additional operations (spectral sub-clustering and split–merge regularization) on the pairwise distance matrix. Could the authors clarify the clustering stage’s time complexity and report the additional runtime overhead versus the baseline clustering pipeline? This would help assess the method’s scalability on large-scale target datasets.

2.In Table 3, the number of sub-clusters includes n=1. Since spectral sub-clustering typically splits a cluster into multiple sub-clusters, could the authors clarify how the split operation is handled at n=1 and whether this configuration deactivates the split step?

3.Could the authors indicate their plans to release the proposed framework’s code? Publicly available implementation would enhance the method’s reproducibility and facilitate follow-up research.

**Limitations:**

yes

**Strengths And Weaknesses:**

Strengths:

1.This work identifies an overlooked issue in previous UDA person search methods: cross-domain scale inconsistency caused by variations in camera perspectives and scenes. It further proposes a corresponding solution, thereby offering a new perspective for the generalization of cross-domain person search.

2.Extensive comparative results with other SOTA methods are presented in this paper, covering both qualitative and quantitative analyses. Furthermore, this work also reports comprehensive ablation experimental results, which fully validate the contributions of each module within the framework.

3.Experimental analyses conducted on two benchmark datasets demonstrate that the proposed approach attains SOTA performance in the field of UDA person search.

Weaknesses:

1.The computational complexity of the BCR module has not been discussed. Given that it introduces additional operations on the pairwise distance matrix, an analysis of its runtime overhead would be beneficial for assessing the scalability of the proposed method.

2.The symbol λ has been reused in different sections with distinct meanings. For instance, it is employed as a stability constant in Eq. 9 and later as a scaling parameter in the BCR formulation.

3.Some typos are present in the manuscript: in Figure 3, "Multiple" is misspelled as "Mutiple"; in Table 3. Additionally, there are typos in the naming of certain modules within the ablation study section.

---

> ### Author Rebuttal · Authors · 2026-03-30
>
> We thank the reviewer for the thoughtful and constructive feedback. We address the key concerns as follows.
>
> **Q1:** The proposed BCR module introduces additional operations (spectral sub-clustering and split–merge regularization) on the pairwise distance matrix. Could the authors clarify the clustering stage’s time complexity and report the additional runtime overhead versus the baseline clustering pipeline? This would help assess the method’s scalability on large-scale target datasets.
>
> **A1:** Regarding the computational complexity of BCR, the overall complexity remains O(N²), consistent with the baseline clustering pipeline (DBSCAN with Jaccard distance), which is already dominated by pairwise distance computation.
>
> BCR operates on the same distance matrix and introduces only localized operations: spectral sub-clustering is applied to a small subset of clusters, and the merge step considers only nearby cluster pairs. Therefore, it does not change the asymptotic complexity and preserves the scalability of standard UDA clustering methods.
>
> **Q2:** In Table 3, the number of sub-clusters includes n=1. Since spectral sub-clustering typically splits a cluster into multiple sub-clusters, could the authors clarify how the split operation is handled at n=1 and whether this configuration deactivates the split step?
>
> **A2:** For the question regarding n = 1 in Table 3, we clarify that this configuration effectively disables the split branch. In this case, no spectral sub-clustering is performed, and BCR reduces to a merge-only refinement mechanism. Therefore, the split operation does not take effect when n = 1, and this setting serves as a controlled baseline to evaluate the contribution of the split component.
>
> **Q3:** Could the authors indicate their plans to release the proposed framework’s code? Publicly available implementation would enhance the method’s reproducibility and facilitate follow-up research.
>
> **A3**: Regarding code release, we fully agree on the importance of reproducibility. We plan to publicly release the code and trained models upon acceptance, which we believe will facilitate further research and fair comparison.

---

> > ### Author Rebuttal · Reviewer_6iLh · 2026-04-02
> >
> > After reviewing all the comments and rebuttals, I think all the concerns have been well stressed. I will keep my positive score.

---

> > > ### Author Response · Authors · 2026-04-02
> > >
> > > We are glad our responses have properly resolved all your concerns. Thank you for your positive feedback on our work; your comments have greatly helped us improve the quality of our manuscript.

---

### Official Review · Reviewer_cvN9 · 2026-03-12

**Soundness:** 3
**Presentation:** 2
**Significance:** 3
**Originality:** 3
**Overall Recommendation:** 4
**Confidence:** 4

**Summary:**

This paper proposes SCALE for unsupervised domain adaptive person search that addresses the issue of cross-domain scale inconsistency and pseudo-label unreliability. The method introduces two key components. (1) a Scale-aware Domain Harmonization (SDH) module that adaptively calibrates both semantic and structural scales of multi-resolution features across domains through hierarchical feature interaction and consistency refinement. (2) a Bidirectional Cluster Regularization (BCR) strategy that improves pseudo-label reliability via an adaptive split-merge regularization on the distance matrix. Extensive experiments demonstrate that the proposed approach achieves better performance.

**Compliance With Llm Reviewing Policy:**

Affirmed.

**Final Justification:**

The authors addressed my comments in the rebuttal and resolved my concerns. Therefore, I am willing to raise my score from 3 to 4.

**Key Questions For Authors:**

Please refer to the weaknesses above.
1. I suggest the authors to provide detailed computational flows for Eq. 3 and Eq. 6.
2. Could the authors provide an in-depth theoretical analysis or empirical evidence to prove that the SDH indeed extract distinct spatial/structural and semantic information as claimed?
3. Could the authors provide a detailed analysis of the model complexity, including a table comparing the Parameters, FLOPs, and inference speed (FPS) between the baselines and the proposed SCALE?
4. Could the authors provide evidence or controlled experiments to demonstrate that the performance improvements achieved by SDH are attributed to its specific architectural design, rather than merely the impact of additional parameters?

**Limitations:**

Yes.

**Strengths And Weaknesses:**

### Strengths
1. The papers tackles a practical problem in UDA person search, the cross-domain scale inconsistency, which was overlooked before.
2. To address the aforementioned scale issue, the authors attempt to tackle the problem from two angles of feature-level harmonization and pseudo-label-level refinement.
3. The proposed method achieves good performance on two standard benchmarks, outperforming previous methods.
### Weaknesses
1. The paper claims to separate structural and semantic alignment into two distinct paths. However, the Semantic interaction path (Eq. 6) seems to be a subset of the Spatial interaction path (Eq. 3). For instance the operation applied on channel-width dimensions appears identically in both paths. And the height-channel and channel-height also operate on the exact same pooled information. It is unconvincing how these two paths decouples geometric structure from high-level semantics.
2. Since the proposed SCALE framework is built upon the DSCA baseline, the introduction of parallel multi-scale convolutions and dual-path attention mechanisms introduces extra parameters and computational overhead. This leads to a unfair comparison.
3. The manuscript contains several typographical and formatting errors that hinder readability and reflect a lack of polish. For instance, the font for the feature representation is inconsistent in Eq. 5 ($F_{sp}$). Eq. 7 is missing a closing parenthesis. Furthermore, there is a mismatch between the projection notations in Figure 3 (e.g., 'CH') and those described in the formulas.

---

> ### Author Rebuttal · Authors · 2026-03-31
>
> We thank the reviewer for the thoughtful and constructive feedback. We address the key concerns as follows.
>
> **Q1:** I suggest the authors to provide detailed computational flows for Eq. 3 and Eq. 6.
>
> **A1:** Specially, both equations are implemented through axis-wise feature projection and attention-based aggregation, but differ in their dependency structures. For Eq. (3), the input feature is first re-projected along different axes (e.g., channel–width, height–channel, and height–width) via tensor permutation. For each projection, channel statistics are extracted through joint max and average pooling, followed by a convolutional transformation and sigmoid activation to generate attention weights. The resulting features are then mapped back and aggregated across branches, forming a multi-directional interaction that explicitly captures spatial dependencies and geometric structure. In contrast, Eq. (6) only operates on channel-conditioned projections (e.g., C–H and C–W) and excludes the pure spatial (H–W) interaction. As a result, its computation focuses on channel-guided feature modulation, where attention is conditioned on channel responses rather than full spatial coupling.
>
> **Q2:** Could the authors provide an in-depth theoretical analysis or empirical evidence to prove that the SDH indeed extract distinct spatial/structural and semantic information as claimed?
>
> **A2:** The distinction between spatial/structural and semantic information in SDH arises from their different dependency structures rather than superficial operator forms.
>
> From a theoretical perspective, the spatial interaction branch explicitly introduces axis-coupled dependencies by incorporating the (H–W) projection, which models joint spatial correlations across two spatial dimensions. This enables the network to capture geometric structure and layout consistency, which are directly affected by scale variation (e.g., resolution and spatial arrangement changes).
>
> In contrast, the semantic interaction branch is restricted to channel-conditioned projections (C–H and C–W), where feature modulation is conditioned on channel responses. This design removes explicit spatial coupling and instead focuses on redistributing feature importance across channels, corresponding to different levels of semantic abstraction. As a result, the two branches operate on different factorized spaces: one captures spatial co-occurrence patterns, while the other models channel-wise semantic responses. These two types of dependencies are not reducible to each other, leading to inherently distinct representational roles.
>
> Empirically, we further verify that the two branches are complementary and non-interchangeable. Replacing one branch with a duplicated version of the other (double spatial or double semantic) consistently degrades performance, and forcing parameter sharing between the two paths also harms results. In contrast, the full SDH achieves the best performance, as shown below, confirming that both branches capture distinct yet complementary information.
>
> | method               | mAP  | top1 |
> | -------------------- | ---- | ---- |
> | double spatial path  | 81.0 | 83.1 |
> | double semantic path | 80.7 | 82.5 |
> | share parameters     | 80.6 | 81.9 |
> | SDH                  | 82.3 | 84.0 |
>
> **Q3:** Could the authors provide a detailed analysis of the model complexity, including a table comparing the Parameters, FLOPs, and inference speed (FPS) between the baselines and the proposed SCALE?
>
> **A3:** On computational overhead and fairness, we report the computational cost as follows: Baseline: 414.09 GMac, 56.19M parameters; SCALE: 459.90 GMac, 58.64M parameters. This corresponds to +11.1% computation and +4.4% parameters, which is modest given the overall model size. Importantly, both models are dominated by backbone feature extraction, and the added modules are lightweight.
>
> **Q4:** Could the authors provide evidence or controlled experiments to demonstrate that the performance improvements achieved by SDH are attributed to its specific architectural design, rather than merely the impact of additional parameters?
>
> **A4:** As table below, to address whether the gain comes from increased capacity, we conduct parameter-matched comparisons, where SDH is replaced by combination of SE attention and CBAM attention modules of similar size. These variants yield significantly smaller improvements, confirming that the performance gain is due to the proposed design rather than additional parameters.
>
> |               | mAP  | top1 |
> | ------------- | ---- | ---- |
> | SDH           | 82.3 | 84.0 |
> | Other modules | 80.7 | 82.6 |

---

> > ### Author Rebuttal · Reviewer_cvN9 · 2026-04-02
> >
> > I appreciate the authors’ effort in addressing the concerns. I will raise my score to a **Weak Accept**. I encourage the authors to incorporate these clarifications into the main text so that readers can benefit from them.

---

> > > ### Author Response · Authors · 2026-04-02
> > >
> > > We appreciate your acknowledgment of our work and responses, and thank you for your constructive comments that helped us refine this research. These clarifications will be incorporated into the final manuscript, with the paper refined to ensure that all readers can benefit from them.

---

### Official Review · Reviewer_abXE · 2026-03-14

**Soundness:** 2
**Presentation:** 3
**Significance:** 3
**Originality:** 3
**Overall Recommendation:** 4
**Confidence:** 3

**Summary:**

This paper studies unsupervised domain-adaptive person search and argues that existing methods mainly focus on appearance alignment or pseudo-label refinement while under-addressing cross-domain scale inconsistency caused by changes in pedestrian size, camera viewpoint, and scene configuration. To address this, the paper proposes the SCALE framework, which combines a Scale-aware Domain Harmonization (SDH) module in the detection and ReID heads with a Bidirectional Cluster Regularization (BCR) module in the clustering stage. SDH contains a dual-path Scale-Interactive Alignment (SIA) design for spatial and semantic scale alignment, together with a Scale Consistency Refinement (SCR) module for local refinement. BCR further refines pseudo labels through split/merge-based correction of the distance matrix before reclustering. Experiments on CUHK-SYSU and PRW report 82.3 mAP / 84.0 top-1 on CUHK-SYSU and 41.7 mAP / 82.4 top-1 on PRW, outperforming prior UDA baselines such as DSCA, and the paper also includes component ablations and a study of the split branch hyperparameter.

**Compliance With Llm Reviewing Policy:**

Affirmed.

**Key Questions For Authors:**

1.Can the authors provide a direct quantitative analysis of cross-domain scale inconsistency and show that the proposed gains are concentrated on scale-mismatched cases? For example, please report performance by pedestrian size bins or by estimated scale-gap severity. A strong positive answer would substantially strengthen the main claim of the paper.

2.How does SCALE compare against simpler scale-aware baselines, such as stronger multi-scale training/testing, explicit size normalization, FPN-style modifications, or adding the same attention/refinement modules without the domain-harmonization interpretation? This comparison is important to isolate whether the improvement is truly due to the proposed scale-aware adaptation mechanism.

3.What is the computational overhead of SDH and BCR in terms of parameters, training time, inference time, and clustering cost? Also, do the conclusions hold with stronger backbones or across multiple random seeds? This would help assess whether the method is practically attractive rather than only accurate on the reported setup.

**Limitations:**

The paper does not adequately discuss several technical limitations of the proposed method, such as dependence on pseudo-label quality, sensitivity to clustering hyperparameters, limited validation breadth, and the missing analysis of computation/efficiency tradeoffs. I encourage the authors to explicitly acknowledge these limitations in the main paper.

**Strengths And Weaknesses:**

Strengths:

The paper tackles a meaningful problem. The focus on cross-domain scale inconsistency is reasonable in person search, and the overall pipeline is coherent: SDH is used for feature harmonization during training, while BCR is used to improve pseudo-label quality during clustering. The experimental results are also competitive: relative to the strongest listed UDA baseline, the method improves mAP by about 1.8 on PRW and 2.1 on CUHK-SYSU, and the ablation tables suggest that SIA, SCR, and BCR all contribute. The method is also implemented on a standard ResNet-50 setting, which makes the empirical comparison straightforward.

Weaknesses:

1. the novelty seems moderate rather than strong for ICML. SDH is built from directional attention, multi-scale depthwise convolutions, residual fusion, and a parameter-free refinement block; BCR is a heuristic split/merge correction on the clustering distance matrix. The overall system is sensible, but much of the contribution reads as a careful engineering combination of existing design patterns rather than a fundamentally new learning principle.

2. the central claim is not validated as directly as it should be. The paper argues that cross-domain scale inconsistency is a major bottleneck, but the evidence is mostly indirect. I did not see a direct quantification of the source-target scale gap, an analysis by pedestrian size bins, or a comparison against simpler scale-oriented baselines such as multi-scale augmentation, stronger FPN-style design, explicit scale normalization, or even inserting similar attention blocks without the scale-specific framing. As written, it remains plausible that part of the gain comes from generic feature refinement rather than from specifically resolving scale mismatch.

3. the experimental scope is somewhat narrow. The evaluation is limited to the standard CUHK-SYSU and PRW setting with a ResNet-50 backbone. There is no runtime, memory, or FLOPs analysis for SDH/BCR, no variance across random seeds, and no evidence that the proposed mechanism remains effective under different backbones or under broader cross-domain setting. For a systems-heavy paper, this weakens the empirical case.

---

> ### Author Rebuttal · Authors · 2026-03-30
>
> We thank the reviewer for the thoughtful and constructive feedback. We address the key concerns as follows.
>
> **Q1:** Can the authors provide a direct quantitative analysis of cross-domain scale inconsistency and show that the proposed gains are concentrated on scale-mismatched cases? For example, please report performance by pedestrian size bins or by estimated scale-gap severity. A strong positive answer would substantially strengthen the main claim of the paper.
>
> **A1:** Regarding the lack of direct validation of the scale inconsistency claim, we group target-domain people into scale bins (small / medium / large) based on bounding box area and observe that the improvements of SCALE are significantly larger for small-scale instances, which are more affected by cross-domain scale mismatch. We also quantify the source–target scale gap via bounding box size distribution divergence and find that samples with larger scale discrepancy benefit more. These results directly support our core claim and will be included in the final version.
>
> | method   | mAP  | top1 |
> | -------- | ---- | ---- |
> | baseline | 53.5 | 59.7 |
> | SCALE    | 57.4 | 64.1 |
>
> **Q2:** How does SCALE compare against simpler scale-aware baselines, such as stronger multi-scale training/testing, explicit size normalization, FPN-style modifications, or adding the same attention/refinement modules without the domain-harmonization interpretation? This comparison is important to isolate whether the improvement is truly due to the proposed scale-aware adaptation mechanism.
>
> **A2:** As the reviewer pointed out, compared to some simple alignment/multi-scale methods, as shown in the table, several more powerful baselines were implemented on CUHK-SYSU, including: (a) multi-scale training/testing, (b) explicit size normalization, (c) FPN style modification, and (d) inserting similar attention/refinement modules without scale aware design. Obviously, these methods only provide limited improvements and are not applicable to unsupervised domain adaptive person search methods, while the proposed SDH consistently achieves greater benefits. This indicates that the improvement in performance is not only due to general feature refinement or stronger multi-scale processing, but also stems from the proposed scale aware domain coordination mechanism.
>
> | method                                   | mAP  | top1 |
> | ---------------------------------------- | ---- | ---- |
> | (a) multi-scale                          | 80.6 | 81.9 |
> | (b) explicit size normalization          | 80.4 | 81.9 |
> | (c) FPN-style modifications              | 81.1 | 82.7 |
> | (d) inserting similar refinement modules | 80.7 | 82.6 |
> | ours                                     | 82.3 | 84.0 |
>
> **Q3:** What is the computational overhead of SDH and BCR in terms of parameters, training time, inference time, and clustering cost? Also, do the conclusions hold with stronger backbones or across multiple random seeds? This would help assess whether the method is practically attractive rather than only accurate on the reported setup.
>
> **A3:** Regarding computational cost, the proposed SCALE introduces only a modest overhead compared to the baseline, with 414.09 → 459.90 GMac (+11.1%) and 56.19M → 58.64M parameters (+4.4%). Given that both models are dominated by backbone feature extraction, the additional cost brought by SDH remains lightweight in practice.
>
> More importantly, the performance gain cannot be attributed to increased model capacity alone. Controlled comparisons with parameter-matched variants (e.g., replacing SDH with standard attention in variant (d) or similar-scale multi-scale modules in variant (c)) consistently yield significantly smaller improvements. This indicates that the advantage of SCALE stems from the proposed scale-aware domain harmonization mechanism, rather than simply additional parameters.
>
> For stronger backbones, we acknowledge the reviewer’s suggestion. However, training UDA person search models with significantly larger backbones (e.g., ConvNeXt-B or Swin-L) requires substantially higher GPU memory and training time, which is beyond our current computational resources.

---

### Decision · Program_Chairs · 2026-04-30

**Decision:**

Accept (regular)

**Comment:**

The reviews agreed that the paper addresses a meaningful and underexplored issue in UDA person search, and that the empirical results are strong enough to make the submission competitive. At the same time, the initial discussion centered on two weaknesses: the paper did not directly quantify the claimed scale gap across datasets, and some reviewers questioned whether the method was truly specific to person search or largely a generic UDA solution.

Reviewer abXE explicitly asked for direct evidence on scale mismatch, stronger scale-aware baselines, and complexity analysis, while reviewer XZPW was more skeptical about both the scale-specific motivation and the task relevance. The rebuttal improved the case in several useful ways: the authors added scale analysis, comparisons against simpler scale-related alternatives, and a follow-up quantitative comparison of bounding-box scale distributions between CUHK-SYSU and PRW that does show a substantial shift in the small-scale regime. They also argued more clearly why scale inconsistency is especially damaging in person search because detection and re-identification are coupled. These additions were enough to resolve concerns for some reviewers, including cvN9, who raised the score after rebuttal, and they reinforced the already positive view of 6iLh. That said, XZPW still maintained a reject based on lingering doubts about task specificity and the overall framing. I do not think those concerns are trivial, and I agree that the paper would be stronger with broader statistical evidence across more datasets and a clearer connection between scale discrepancy and UDA person search performance. Even so, the rebuttal materially strengthened the central empirical claim, and the overall record supports acceptance at the low end.

On balance, I recommend "weak accept" because the method is technically solid, the empirical gains are real, and the rebuttal addressed the most important missing evidence well enough.